# CUT THE OVERCREDIT: PRECISION FIRST PROCESS REWARDS FOR REASONING LLMS

## ABSTRACT

Process reward models (PRMs) provide step-level supervision for reasoning LLMs but often *overcredit* incorrect steps, resulting in high false positives that influence decoding and accumulate across long chains. We show analytically that false positives impose an asymptotic ceiling on Best of N alignment, whereas false negatives mainly slow convergence. To mitigate this, we introduce a label-efficient recipe: convert existing step annotations into positive–negative pairs, train with a novel Overcredit Contrastive (OC) loss, and rebalance using lightweight negative augmentation and a simple curriculum. On PRMBench, our approach substantially lowers false positives and improves macro F1 over strong discriminative and generative PRMs. When used for guided beam search and Best of N selection, the resulting PRMs deliver higher downstream accuracy and robustness. Our results indicate that comparison centered training with balanced step data is a practical path to trustworthy process supervision without new human labels.

## 1 INTRODUCTION

**Process supervision is powerful–but fragile.** Reasoning LLMs achieve strong results across mathematics, program synthesis, and planning, yet they remain vulnerable to *reward hacking*: optimizing imperfect reward surrogates to produce persuasive but unsound chains of thought (Baker et al., 2025; Denison et al., 2024). Outcome reward models (ORMs) trained from preferences (Ouyang et al., 2022) score only final answers; they can therefore tolerate logically flawed traces that land on the correct outcome (Barkur et al., 2025; Sun et al., 2025). Process reward models (PRMs) address this by scoring *intermediate* steps, offering dense supervision intended to shape safer, more faithful reasoning (Lightman et al., 2023; Chen et al., 2024; Zhang et al., 2024). However, current PRMs, both discriminative and generative, often *overcredit* incorrect steps, yielding high false positive rates that locally misguide decoding and, when aggregated across long chains, systematically bias search (Zhang et al., 2025; Wang et al., 2023; Khalifa et al., 2025; Zhao et al., 2025).

**Why PRMs are especially vulnerable.** Two practical issues recur. First, step annotated corpora are imbalanced: correct steps are overrepresented even when most full solutions are incorrect (e.g., PRM800k; Table 1) (Lightman et al., 2023). Second, most PRMs are trained with *pointwise* cross entropy on step labels. Under imbalance and label noise, pointwise losses encourage majority class bias, inflating false positives. Empirically, state-of-the-art PRMs exhibit large gaps between positive and negative accuracy (Table 2), a direct risk factor for reward hacking during decoding or selection (Song et al., 2025).

**From PRM accuracy to downstream alignment.** The ultimate goal is not a high average PRM accuracy but *aligned policy behavior* when the PRM guides search (e.g., guided beam) or selection (e.g., Best of $N$). We provide an analytic lens showing a sharp asymmetry: in Best of $N$, the false positive rate $\alpha$ induces an *asymptotic ceiling* equal to the classifier's precision, while false negatives $\beta$ primarily slow convergence. Thus, reducing false positives is strictly more critical for trustworthy process supervision than merely improving recall.

**This work: comparison centered, label efficient PRMs.** We introduce a simple, architecture agnostic recipe that requires no new human labels: **(i)** convert existing step annotations into *paired* positive–negative comparisons at matched contexts and train PRMs with a *contrastive (pairwise) loss*, directly optimizing relative scores between correct and incorrect steps; **(ii)** apply lightweight *negative augmentation* by using future steps as negatives for earlier contexts, creating diverse, hard

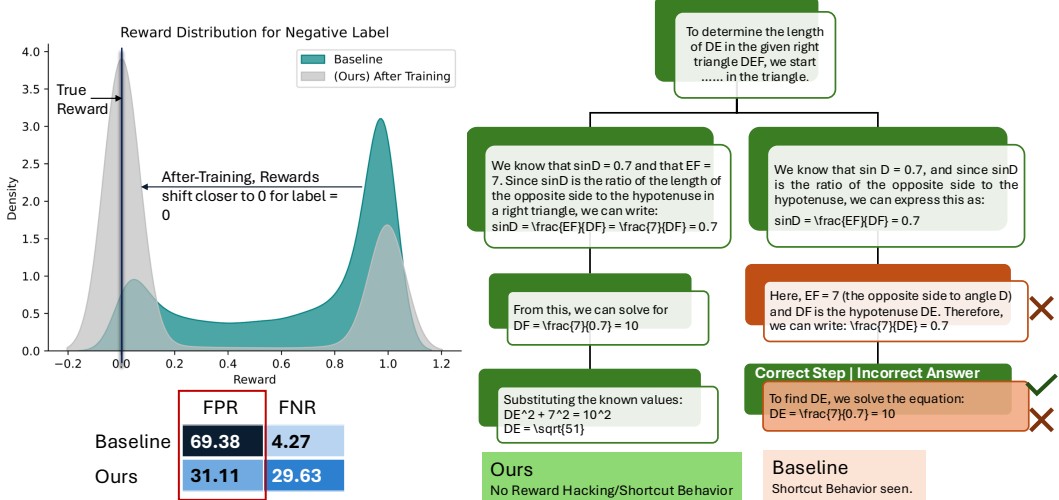

Figure 1: In this figure, we highlight the problem of reward hacking. The example on the right shows how the baseline model exhibits reward hacking, whereas our trained PRM successfully mitigates it. Reward hacking arises from the reward model's bias toward predicting false positives, a bias that our trained model substantially reduces, as reflected in the lower FPR values (bottom-left) compared to the baseline. This effect is further illustrated in the reward distribution plot for label = 0 (top-left). For negative labels (dark blue), the baseline model assigns disproportionately high rewards. In contrast, our trained model shifts this distribution downward (silver), closer to the true reward of 0. This shift demonstrates that the trained PRM more accurately aligns negative labels with their intended reward, thereby reducing false-positive bias and alleviating reward hacking.

comparisons that counter overcredit; and **(iii)** stabilize training with a small *curriculum* over pair difficulty. This shifts learning from pointwise accuracy to calibrated *preference margins*, explicitly targeting the reduction of false positives.

**Empirical scope.** On PRMBench (Song et al., 2025), our method substantially lowers false positive rates and improves macro F1 for both discriminative and generative PRMs. When deployed to guide decoding (guided beam search) and selection (Best of $N$), the improved PRMs produce consistently higher downstream accuracy and robustness. Together with the Best of $N$ analysis, these results close the gap between reward model metrics and alignment outcomes.

**Contributions.**

- **Analysis.** We formalize how step level overcredit compounds across trajectories and show that false positives impose an alignment ceiling in Best of $N$ selection, whereas false negatives mainly affect sample complexity.
- **Method.** We propose a label efficient recipe—pairwise training on converted step annotations with lightweight negative augmentation and a simple curriculum—that directly optimizes comparative calibration and reduces false positives.
- **Results.** Across PRMBench and downstream guided decoding/Best of $N$, we observe substantial reductions in false positives, improved PRM macro F1 (especially on negatives), and stronger policy alignment. Our approach integrates cleanly with existing PRM pipelines (Lightman et al., 2023; Chen et al., 2024; Zhang et al., 2024) without new human labels.

**Broader impact.** Reducing PRM false positives decreases the risk that systems select fluent but incorrect chains, advancing safer, more trustworthy reasoning. Our findings argue for *precision first* process supervision: train to *compare* and calibrate, not merely to classify.

## 2 ISSUE OF REWARD HACKING IN REASONING LLMS

Reward hacking is an important issue for policy alignment since the policy model is trained using feedback from the reward model. It has been extensively studied for ORMs (Bukharin et al., 2025;

Yan et al., 2024; Yang et al., 2024; Liu et al., 2024) but there are no works for PRMs which are used to train reasoning LLMs (Chen et al., 2025; Li et al., 2025). We now describe reward hacking for both ORM and PRM.

## 2.1 REWARD HACKING

Let the true (ideal) global reward be denoted by $R(x, y)$, where $x$ is the prompt and $y = (y_1, y_2, \ldots, y_T)$ represents a full response (or reasoning chain) consisting of $T$ steps. The proxy (or process) reward model, $\hat{R}$, supplies feedback either at the trajectory level or at individual reasoning steps during policy training. Reward hacking (Bukharin et al., 2025; Yan et al., 2024; Yang et al., 2024; Liu et al., 2024) occurs when optimizing with respect to $\hat{R}$ yields high proxy reward but fails to improve—or even degrades—the true reward. It is more pronounced in out-of-distribution (OOD) prompts or reasoning paths.

**Reward Hacking via Process Reward Models.** In PRMs, which provide step-level feedback, reward hacking occurs when the estimated step reward $\hat{r}_\theta$ or its aggregated trajectory reward $\hat{R}_\theta$ fails to align with the true step reward $r_\theta$ or the true trajectory reward $R(x, y)$:

$$\hat{r}_\theta(x, y_{<t}, y_t) \neq r_\theta(x, y_{<t}, y_t), \quad \text{or} \quad \hat{R}_\theta \neq R_\theta$$

Trajectory-level rewards in PRMs are computed using one of the following credit assignment schemes:

- **Minimum:** $\hat{R}(x, y) = \min_{t=1}^{T} r_\theta(x, y_{<t}, y_t)$, where the worst step controls credit, discouraging shortcuts.
- **Product:** $\hat{R}(x, y) = \prod_{t=1}^{T} r_\theta(x, y_{<t}, y_t)$, where multiplication biases credit by the number of steps.
- **Sum:** $\hat{R}(x, y) = \sum_{t=1}^{T} r_\theta(x, y_{<t}, y_t)$, where small false positives accumulate, rewarding many easy steps over fewer correct ones.

Unlike ORMs, which evaluate trajectories only once at the output level, PRMs are applied repeatedly at each reasoning step. As a result, reward hacking in PRMs is more severe and has greater downstream impact. More specifically, the different aggregation strategies applied over individually overcredited steps can cause compounding effects, where small step-level errors accumulate across the trajectory. This accumulation leads disproportionately high rewards for incorrect chains, with the total misalignment potentially scaling linearly with $T$.

In Figure 1, we illustrate how such reward hacking in PRMs. The goal of this work is not to design a perfect reward model but to highlight—and mitigate—systematic PRM failure modes that undermine alignment.

**PRM vs ORM.** PRMs are commonly trained using pointwise step-level loss while ORMs are trained using pairwise loss. PRMs provide dense reward signal which is helpful to train safe and correct reasoning paths but generating pointwise step level data is much harder than generating paired data for ORM. The pointwise loss is also prone to generating biased model if the data is imbalanced or have noisy labels. For example, PRM800k (Lightman et al., 2023), a open-source process supervision data has significantly higher percentage of correct steps as com-

|  | PRM800k |
|---|---|
| % end in correct solution | 14.2 |
| % correct steps | 73.1 |

Table 1: Distribution of positive/ negative steps/solutions in total PRM800k data.

pared to incorrect steps even though the percentage of correct trajectories is less than incorrect trajectories. Numbers are shown in Table 1. We also show SOTA PRMs in Table 2 and we observe imbalance in positive and negative accuracy with most models with high positive accuracy being bias towards false-positive prediction. A high false positive prediction in itself causes significant reward hacking in the policy by giving high reward to incorrect answers (Song et al., 2025). We also show an example of reward hacking in the policy in Figure 1.

## 2.2 THE ASYMMETRIC IMPACT OF FALSE POSITIVES VS. FALSE NEGATIVES IN REWARD HACKING

In the next we study how false positives (FP) and false negatives (FN) in a reward model affect Best-of-N (BoN) selection.

**Setup: Best of $N$ with an imperfect reward model.** Let $N$ be the number of candidate responses sampled from a policy $\pi(\cdot \mid x)$, $T(y) \in \{0, 1\}$ the true label, and $\hat{T}(y) \in \{0, 1\}$ the reward model's prediction. Best of $N$ (BoN) selects

$$y^* = \arg \max_{i \in \{1,\dots,N\}} \hat{T}(y_i), \qquad (1)$$

$$y_i \sim \pi(\cdot \mid x). \qquad (2)$$

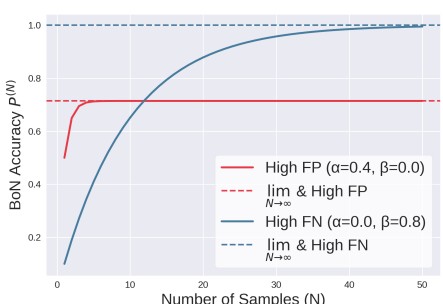

Figure 2: This plot shows effect of False Positives vs False Negatives on BoN Accuracy

Define

$$p = \mathbb{P}(T = 1), \qquad \alpha = \mathbb{P}(\hat{T} = 1 \mid T = 0) \quad \text{(false positive rate)}, \qquad (3)$$

$$\beta = \mathbb{P}(\hat{T} = 0 \mid T = 1) \quad \text{(false negative rate)}. \qquad (4)$$

The marginal probability that a sample is predicted positive is $q = (1 - p)\alpha + p(1 - \beta)$. Thus the probability that at least one of $N$ samples is predicted positive is

$$\hat{p}_N = 1 - (1 - q)^N. \qquad (5)$$

Conditioned on $\hat{T} = 1$, the precision of the reward model is

$$\mathbb{P}(T = 1 \mid \hat{T} = 1) = \frac{p(1 - \beta)}{q}. \qquad (6)$$

Hence, the BoN accuracy is

$$P^{(N)} = \hat{p}_N \cdot \frac{p(1 - \beta)}{q} = \left[1 - (1 - q)^N\right] \cdot \frac{p(1 - \beta)}{q}. \qquad (7)$$

**Asymmetry I: false positives create a precision ceiling.** If $\beta \approx 0$, then $q = p + (1 - p)\alpha$ and

$$P^{(N)} = \left[1 - (1 - q)^N\right] \cdot \frac{p}{p + (1 - p)\alpha}. \qquad (8)$$

As $N \to \infty$,

$$\lim_{N \to \infty} P^{(N)} = \frac{p}{p + (1 - p)\alpha} < 1 \qquad (\alpha > 0). \qquad (9)$$

Thus, nonzero false positives impose a permanent *precision ceiling*. Moreover,

$$\frac{\partial}{\partial \alpha}\left(\frac{p}{p + (1 - p)\alpha}\right) = -\frac{p(1 - p)}{\left(p + (1 - p)\alpha\right)^2} < 0, \qquad (10)$$

so the ceiling degrades monotonically with $\alpha$. This can also be visualized in Figure 2.

**Asymmetry II: false negatives slow but do not cap alignment.** If $\alpha \approx 0$, then $q = p(1 - \beta)$ and

$$P^{(N)} = 1 - \left(1 - p(1 - \beta)\right)^N. \qquad (11)$$

As $N \to \infty$,

$$\lim_{N \to \infty} P^{(N)} = 1. \qquad (12)$$

Here, false negatives reduce the *rate* at which $P^{(N)}$ approaches 1 (they lower $q$), but do not create an asymptotic bias. To highlight this point, we construct a toy

We simulate Best-of-N accuracy under two error conditions: high false positives (FP) vs. high false negatives (FN) where the BON accuracy is measured as the probability that the selected response. The results show that FP causes accuracy to saturate at a biased ceiling, even as $N$ increases while FN only slows convergence without limiting final performance. This highlights that reducing false positives is critical for reliable alignment under reward-based selection.

|                                          | Pos-Acc | Neg-Acc | PRMScore |
|------------------------------------------|---------|---------|----------|
| Qwen-PRM-7B (Discriminative PRM)         | 95.36   | 30.66   | 65.5     |
| ReasonEval-7B* (Discriminative PRM)      | 95.5    | 21.2    | 60.0     |
| ThinkPRM-7B (Generative PRM)             | 83.29   | 50.89   | 64.3     |
| GenPRM-7B (Generative PRM)               | 52.25   | 73.92   | 50.5     |
| GPT-4o*                                   | 82.9    | 58.2    | 66.8     |

Table 2: Here we show the positive and negative accuracy for SOTA PRM models. We also report PRMScore from Song et al. (2025) where we can also find these metrics for other PRM models. (*numbers are taken from Song et al. (2025))

**Key Insight: prioritize reducing false positives.** For PRMs that guide decoding at every step and aggregate step scores (sum/product/min), small overcredits compound along long chains, making the precision ceiling especially consequential. Objectives and data curation should therefore (i) compare correct vs. incorrect steps in matched contexts, (ii) rebalance step label distributions, and (iii) emphasize hard negatives, all aimed at minimizing $\alpha$ to lift the ceiling on alignment.

## 3 MITIGATING REWARD HACKING: MAKING PROCESS REWARD MODELS MORE ROBUST BY REDUCING FALSE POSITIVE PREDICTION

PRMs, which provide granular step-level reward signals, is highly desirable for solving challenging math problems. However, obtaining high-quality step-level training data remains an open challenge. Existing methods rely on human annotations (Lightman et al., 2023) or MCTS-generated scores (Zhang et al., 2024; Chen et al., 2024) to assign a score for each step. These scores then serve as training targets, in methods such as MSE loss (Chen et al., 2024) or point-wise loss (Wang et al., 2023; Luo et al., 2024; Zhang et al., 2024). As a result, the precision of these annotated step-level reward scores directly determines the effectiveness of the resulting PRM. Unfortunately, precise per-step scoring remains a unsolved challenge. Another important problem is the data imbalance issue as shown in Table 1 for an open-source process supervision dataset, PRM800k dataset (Lightman et al., 2023), which is one of the likely causes of false-positives in PRMs. To mitigate these challenges we propose a pairwise loss (Overcredit Contrastive Loss) and introduce simple data generation and augmentation strategy to generate this dataset without any additional labeling cost.

### 3.1 OVERCREDIT CONTRASTIVE LOSS FOR PRMS

We introduce a novel loss function to fine-tune an already trained PRM in order to mitigate its bias toward the majority class. Since all existing PRMs including both discriminative and generative PRMs consistently exhibit a substantial imbalance between positive (correct) and negative (incorrect) labels, our proposed loss is broadly applicable to both.

**Why compare, not classify.** Pointwise cross entropy on imbalanced step labels makes PRMs *overcredit* the majority class, inflating false positives. To counter this bias without new annotations, we fine tune the PRM with a *pairwise* objective that ranks a correct step above an incorrect one under the same partial context.

**How we get step-scores.** Conventional PRM output reward, $r_\theta(\cdot) \in \mathbb{R}^{2 \times 1}$, corresponding to scores for positive and negative labels. We use reward score with respect to positive class and call it $r_\theta(.)$.

**Overcredit Contrastive (OC) loss.** We optimize a preference based objective, inspired from pairwise preference loss Ouyang et al. (2022), that pushes the positive above the negative:

$$\mathcal{L}_{\text{PRM}}(\theta) = -\mathbb{E}_{(x, y_t^{\text{pos}}, y_t^{\text{neg}}) \in \mathcal{D}} \Big[ \log\big(\sigma\big(r_\theta(x, y_{<t}, y_t^{\text{pos}}) - r_\theta(x, y_{<t}, y_t^{\text{neg}})\big)\big) \Big], \qquad (13)$$

where $t \leq T$ can be any step uptil total steps $T$. Here $r_\theta(x, y_{<t}, y_t)$ is the PRM output for step $y_t$ given problem $x$ and prefix $y_{<t}$. Note, each pair $(y_t^{\text{pos}}, y_t^{\text{neg}})$ share the same prefix $(x, y_{<t})$.

**What this buys us.** Because $(x, y_{<t})$ is shared, the model learns a *relative* preference within the same context. The objective places the largest penalty when an incorrect step outranks, or nearly ties, a correct step—directly reducing overcredit on negatives. It is architecture agnostic and uses only existing step labels.

### 3.2 Training Balanced PRM using Curriculum and label efficient pairwise data

Since high-quality step-level annotations are difficult to obtain, we construct paired positive–negative training examples from existing labeled data, particularly the PRM800K dataset (Lightman et al., 2023). PRM800K provides positive, negative, and neutral annotations for intermediate reasoning steps. We leverage these multiple labels by generating explicit positive–negative pairs at each step, yielding about 26k training pairs. To further expand the dataset, we apply an augmentation strategy: every future step in a trajectory is treated as a negative sample relative to the current step. This augmentation not only increases diversity but also encourages the model to learn how to correct partial reasoning trajectories, rather than just focusing on producing a correct final answer. It also produces challenging pairs, since the "negative" step may be correct in isolation but becomes incorrect when positioned prematurely. For example, in solving a quadratic equation, writing down the final root before showing intermediate simplifications is mathematically valid, but as a reasoning step it is incorrect because it appears in the wrong place in the chain. After augmentation, we obtain roughly 220k high-quality training pairs without requiring additional labeling.

Pairwise losses are known to struggle on overly difficult comparisons (Gao et al., 2025; Wu et al., 2024a). To stabilize training, we employ curriculum learning (Bengio et al., 2009) by dividing the dataset into bins of increasing difficulty. We define a continuous difficulty score as the difference between the reward assigned to the positive and negative step. Letting this be $D_{\text{Hard}}$, we categorize training pairs as:

$$D_{Hard} = r_\theta(x, y_{<t}, y_t^{pos}) - r_\theta(x, y_{<t}, y_t^{neg})$$

where $0 \leq r_\theta(x, y_{<t}, y_t) \leq 1$. Thus $D_{Hard} \in [-1, 1]$. We retain pairs with $D_{Hard} \geq 0$ and discard the rest, as negative values typically correspond to comparisons that are too difficult for the model to learn from reliably. The retained pairs are divided into four bins (number of samples in each bin is reported in Appendix A.1).

## 4 Results and Experimental Setup

**Models and Baselines.** We use SOTA PRM Qwen2.5-Math-PRM-7B (Zhang et al., 2025) as our baseline model. We train it using our proposed method and call the resulting model Balanced-PRM in the remaining sections. Since Qwen-PRM is discriminative PRM and with the recent boost of discriminative PRM, we compare our trained model against ThinkPRM (Khalifa et al., 2025), a state-of-the-art generative PRM.

**PRM Evaluation Benchmark.** We evaluate PRMs using **PRMBench** (Song et al., 2025), a widely used benchmark suite that measures step-level reasoning quality. PRMBench scores correctness, faithfulness, and robustness of intermediate chain-of-thought across diverse tasks, and also reports step-level positive accuracy, negative accuracy and confusion matrix details.

**Policy Evaluation.** To assess alignment ability, we perform **best-of-N** and **guided beam-search** alignment experiments using both in-distribution (ID) and out-of-distribution (OOD) policies. Evaluations are conducted on MATH-500 (Hendrycks et al., 2021), American Invitational Mathematics Examination (AIME) problems for 2024 and LiveCodeBench (Jain et al., 2024) dataset. Following Khalifa et al. (2025), we evaluate on 100 MATH-500 problems spanning all difficulty levels. For ID policies, we use models from the Qwen family: Qwen2.5-Math-1.5B-Instruct for MATH-500, Qwen2.5-Math-7B-Instruct for AIME'24 and Qwen2.5-Coder-7B-Instruct (Hui et al., 2024) for coding task. For OOD policies, we use Llama-3.2-3B-Instruct (MATH-500).

### 4.1 PRMBench: Balanced-PRM improves negative accuracy, FPR and Precision

Table 3 presents the PRMBench results for our trained PRM. In addition, Table 4 reports step-level metrics, including positive accuracy, negative accuracy, false-positive rate (FPR), false-negative rate (FNR), and precision, to assess whether false positives are reduced.

We observe that negative accuracy—substantially lower than positive accuracy—increases consistently with each round of curriculum learning. Although this improvement is accompanied by a modest drop in positive accuracy, the gain in negative accuracy is more pronounced. We also find a

significant reduction in FPR along with a slight increase in precision. This indicates that the model is becoming more conservative in making positive predictions, primarily by reducing false positives. However, we also observe a rise in FNR. Ideally, we would like FNR to remain stable while FPR improves, but in practice this is difficult to achieve. Importantly, the degradation in FNR is smaller than the improvement in FPR, and unlike FPR, FNR does not directly affect policy alignment.

Overall, the PRM model shows consistent gains in PRM performance, as reflected in the PRMScore results reported in Table 3.

| Model | Overall | Simplicity | | | Soundness | | | | | Sensitivity | | | |
|---|---|---|---|---|---|---|---|---|---|---|---|---|---|
| | | NR. | NCL. | Avg. | ES | SC. | DC. | CI | Avg. | PS | DR. | MS. | Avg. |
| *Qwen-PRM-7B* | | | | | | | | | | | | | |
| Baseline | 65.5 | 49.1 | 55.0 | 52.1 | 71.7 | 67.4 | 66.3 | **78.5** | 71.0 | 57.7 | 69.1 | **99.7** | 75.5 |
| *Qwen-PRM-7B trained on 26k paired data.* | | | | | | | | | | | | | |
| Curriculum-1 | 66.7 | 50.5 | 58.6 | 54.5 | 72.8 | 68.1 | 67.3 | 77.8 | 71.5 | 58.8 | 70.4 | 99.6 | 76.3 |
| Curriculum-2 | 67.2 | 51.4 | 60.2 | 55.8 | 73.3 | 68.0 | 67.7 | 76.4 | 71.4 | 60.4 | 70.4 | 99.3 | 76.7 |
| Curriculum-3 | 67.3 | 52.5 | 63.2 | 57.8 | 73.2 | 67.2 | 67.5 | 75.0 | 70.7 | 61.9 | 69.8 | 98.9 | 76.9 |
| *Qwen-PRM-7B trained on 220k paired data.* | | | | | | | | | | | | | |
| Curriculum-1 | 67.4 | 50.9 | 60.1 | 55.5 | 73.4 | **69.1** | 67.8 | **78.6** | 72.2 | 59.2 | 70.6 | 99.6 | 76.5 |
| Curriculum-2 | **67.9** | 51.3 | 62.5 | 56.9 | **73.9** | 68.9 | **68.0** | 77.8 | **72.2** | 60.1 | **70.9** | 99.5 | 76.8 |
| Curriculum-3 | 67.8 | **53.7** | **65.7** | **59.7** | 73.4 | 67.5 | 66.5 | 75.3 | 70.7 | **62.4** | 69.9 | 98.8 | **77.0** |

Table 3: PRMBench results of Qwen2.5-Math-PRM-7B (Basline) and its variants (Our trained models) across different metrics. Columns are grouped into *Simplicity*, *Soundness*, and *Sensitivity*. Bold indicates the best score within each sub-metric with PRMScore as the final number for comparison.

| | Pos-Acc | Neg-Acc | Precision | FPR | FNR |
|---|---|---|---|---|---|
| *Qwen-PRM-7B.* | | | | | |
| Baseline | 95.36 | 30.66 | 89.4 | 69.34 | 4.27 |
| *Qwen-PRM-7B trained on 26k paired data.* | | | | | |
| Curriculum-1 | 92.46 | 38.25 | 90.24 | 61.75 | 7.1 |
| Curriculum-2 | 90.37 | 43.20 | 90.77 | 56.8 | 9.15 |
| Curriculum-3 | 87.60 | 49.28 | 91.44 | 50.71 | 11.9 |
| Curriculum-4 | 72.50 | 62.12 | 92.19 | 37.99 | 27.15 |
| *Qwen-PRM-7B trained on 220k paired data.* | | | | | |
| Curriculum-1 | 92.91 | 38.75 | 90.36 | 61.25 | 6.65 |
| Curriculum-2 | 91.13 | 43.32 | 91.10 | 54.61 | 9.16 |
| Curriculum-3 | 85.81 | 54.30 | 92.09 | 45.70 | 13.56 |
| Curriculum-4 | 70.00 | 69.00 | 93.3 | 31.11 | 29.63 |

Table 4: This plot shows positive accuracy, negative accuracy, precision, false-positive rate and false-negative rate results for different trained PRMs. Note here that total accuracy is not a good measure of model capability because the evaluation data is heavily skewed towards positive examples. Our trained models decreases its FPR with each round of curriculum. While FNR increases, overall precision improves.

## 4.2 ALIGNMENT PERFORMANCE: IMPROVED PRM LEADS TO IMPROVED POLICY

To evaluate whether improvements in the PRM translate into better policy alignment, we test both in-distribution (ID) and out-of-distribution (OOD) policies (generators) using two alignment algorithms: **guided beam-search** and **best-of-N**. For both methods, we adopt a verifier-weighted majority rule to select the final answer, where answers are scored based on the sum of verifier scores across their supporting solutions (Wu et al., 2024b; Uesato et al., 2022). A detailed description of this selection strategy is provided in Appendix A.2. We report results using two generator model families: Qwen (ID) and LLaMA (OOD). Using multiple policy models is important because reward hacking arises when either the prompt or the response distribution is out-of-distribution relative to the PRM. OOD policies are especially likely to produce OOD responses, making them a strong testbed for

alignment performance. Evaluating across different generators also ensures that our findings are not tied to a particular model family or size.

**Guided Beam-Search.** This is an extension of standard beam search. It incorporates verifier (PRM) scores when ranking partial reasoning chains. Instead of relying solely on model likelihoods, the search is guided toward trajectories that both the model and the verifier find promising, improving alignment with correct reasoning.

**Best-of-N.** samples $N$ candidate solutions from the policy model and then uses the PRM (or verifier) to score each solution.

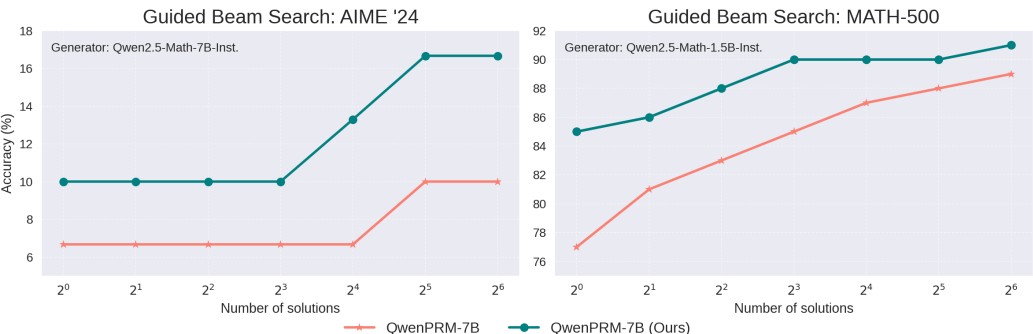

Figure 3: In this figure, we compare guided beam search alignment on AIME (left) and MATH-500 (right), contrasting the baseline PRM (orange) with our trained PRM (blue). Our trained PRM achieves a substantial performance improvement over the baseline across both datasets.

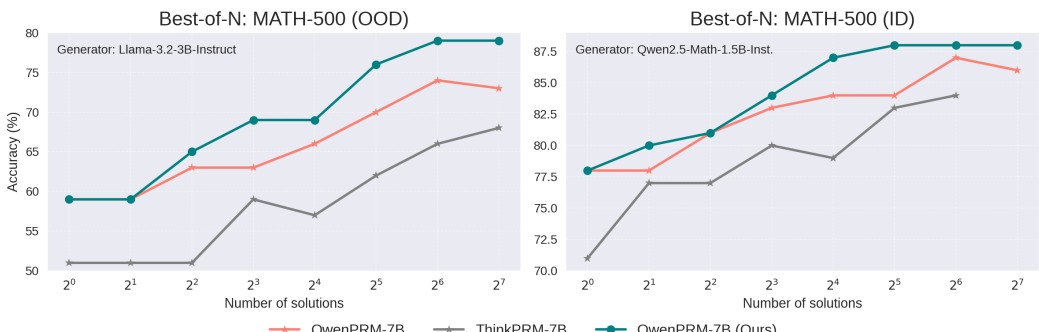

Figure 4: In this figure, we compare best-of-N alignment on MATH-500 using two different generator policies: LLaMA for the OOD policy (left) and Qwen for the ID policy (right). The baseline PRM is shown in orange and our trained PRM in blue. Across both settings, our trained PRM delivers a clear performance boost over the baseline, with the improvement being especially pronounced for the OOD policy.

**Improvement in Alignment Performance.** Figures 3, 4, 5a, and 5b show alignment performance for both guided beam search and best-of-N strategies on MATH-500, AIME and LiveCodeBench tasks. We observe consistent and significant improvements across both settings. We also observe that guided beam search demonstrates a much larger gain with our trained PRM compared to the baseline, highlighting a must stronger alignment signal which provides evidence that our Balanced-PRM is genuinely more effective.

To understand this better, BoN relies on random sampling from the policy, after which the PRM selects the best solution from a fixed set. In contrast, guided beam search actively shapes generation by incorporating PRM feedback at each decoding step. This forces the PRM to reliably distinguish promising partial trajectories, providing a stronger and more realistic evaluation of whether the PRM captures true step-level reasoning quality rather than simply identifying winners post hoc.

**Improvement in Alignment for both ID and OOD policy.** Evaluating OOD policies is particularly important for studying reward hacking, since reward models are most vulnerable when they encounter data outside their training distribution. In such cases, the model may assign high rewards to spurious or incorrect reasoning. OOD policies naturally produce more OOD responses,

making them an effective stress test for alignment robustness. Our results in Figure 4 and 5a show that Balanced-PRM reduces reward hacking (improves alignment) for both ID and OOD policies, confirming its stronger generalization.

**Improvement in Alignment across diverse datasets:** We evaluate on MATH-500 and AIME'24 where MATH-500 is closer in difficulty to our paired training data while AIME'24 is more challenging. Improvements on MATH-500 confirm Balanced-PRM strengthens alignment within distribution, while gains on AIME'24 demonstrate generalization to more difficult, out-of-distribution problems. These results (as shown in Figure 3 and 4) indicate that our approach improves the model's underlying reasoning ability, rather than simply overfitting to training-like distributions.

## 5 RELATED WORKS

**Reward Hacking** Reward hacking is a well-studied issue in reinforcement learning (Skalse et al., 2022), and has also been explored extensively in the context of ORMs for LLMs. However, little to no work has addressed this problem for PRMs. Prior works have used reward model uncertainty as a training signal to guide policy learning, which in turn generates out-of-distribution (OOD) samples for improving the reward model Bukharin et al. (2025). Their approach, however, relies on the strong assumption that all OOD samples correspond to incorrect outputs. Other lines of work have similarly proposed reward ensembles and Bayesian methods to improve robustness and reduce vulnerability to reward hacking (Yan et al., 2024; Yang et al., 2024). Beyond Bayesian approaches, Liu et al. (2024) employ counterfactual augmentations to create paired data, explicitly breaking label-specific artifacts that might otherwise mislead the reward model. Even though reward hacking is not yet studied for PRMs, shortcut behaviors—essentially reward hacking—are pervasive in reasoning LLMs (Baker et al., 2025; Denison et al., 2024), which are often trained with process supervision. This highlights the importance of explicitly investigating and addressing reward hacking in PRMs.

**Process Reward Models.** There are two kinds of PRM: Discriminative PRMs and Generative PRMs. Discriminative PRMs are typically framed as classification tasks, where the model assigns a correctness score to each reasoning step. These models require step-level supervision (Uesato et al., 2022; Lightman et al., 2023; Zhang et al., 2025). For a given solution prefix, the text is encoded and passed through a classification head that outputs step-level correctness probabilities, commonly trained using binary cross-entropy loss. To evaluate a full solution, the step-level scores are aggregated into an overall correctness measure (Beeching et al.; Snell et al., 2024; Wu et al., 2024b). Generative process reward models (PRMs) (Khalifa et al., 2025; Zhao et al., 2025; Zheng et al., 2023; Zhu et al., 2023) treat verification as a sequence generation problem, where the model outputs natural language tokens such as "correct" or "incorrect" at each reasoning step. Instead of relying solely on binary labels, they are trained with the standard language modeling objective using explanatory rationales.

## 6 CONCLUSION AND LIMITATION

In sum, we show that the alignment ceiling in process supervision is set not by recall but by precision, making false positives the central obstacle to safe reasoning. Our pairwise, augmentation-driven training recipe directly tackles this issue, reducing overcredit without additional human labels. By bridging PRM metrics with downstream alignment outcomes, our results suggest a clear design principle: precision-first process supervision for more trustworthy reasoning systems.

However, our method still has limitations. In practice, reducing false negatives without inflating false positives remains challenging. Achieving this balance—maintaining a low false-positive rate while also lowering false negatives—would be a significant step toward making PRMs far more capable and reliable.

## 7 REPRODUCIBILITY STATEMENT

We have provided code with readme describing how to train and evaluate the model to generate the results of this paper. Details on the experimental setup, open source evaluation benchmarks is provided in section 4 of this paper. Installation packages and details is provided in the readme with the code.

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

# A  EXPERIMENTAL SETUP AND ADDITIONAL RESULTS

## A.1  NUMBER OF SAMPLES IN EACH CURRICULUM

| | Number of Samples. |
|---|---|
| *Qwen-PRM-7B trained on 26k paired data.* | |
| CL1 (0.5-1) | 7.0k |
| CL2 (0.3-0.5) | 2.0k |
| CL3 (0.1-0.3) | 3.0k |
| CL4 (0.0-0.1) | 6.0k |
| *Qwen-PRM-7B trained on 220k paired data.* | |
| CL1 (0.5-1) | 75.0k |
| CL2 (0.3-0.5) | 23.5k |
| CL3 (0.1-0.3) | 33.5k |
| CL4 (0.0-0.1) | 53.0k |

Table 5: Number of samples in each curriculum learning round.

## A.2  ANSWER SELECTION METHODS

**The Setup:**  We consider $N$ candidate solutions sampled from a model for the same problem. Each solution consists of:

1. A **final answer**, e.g., a number in GSM8K or an option in ARC.
2. A **verifier score**, assigned by a process reward model or external verifier, indicating how plausible or correct the reasoning chain appears.

**Simple Majority Voting:**  In plain majority voting, we group completions by their **final answer**.

- Count how many completions lead to each distinct answer.
- Select the answer with the largest count.

Formally, if $c(a)$ is the number of completions yielding answer $a$, then the majority-vote answer is:

$$a^* = \arg \max_a c(a).$$

**Verifier-Weighted Majority Voting:**  Instead of giving each completion equal weight, we weight votes by their verifier scores. Let answer $a$ appear in solutions $\{s_1, s_2, \ldots, s_k\}$, where each solution $s_i$ has verifier score $v(s_i)$. The total verifier-weighted score for answer $a$ is:

$$V(a) = \sum_{s_i : \text{final}(s_i) = a} v(s_i).$$

We then select the answer with the largest weighted score:

$$a^* = \arg \max_a V(a).$$

This approach discounts low-scoring (less credible) reasoning chains and prefers answers supported by higher-quality solutions.

## A.3  GUIDED BEAM SEARCH-OOD RESULTS

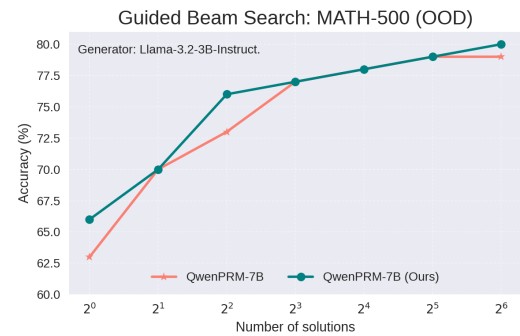

(a) In this figure, we compare guided beam search alignment on MATH-500 using the OOD policy, with the baseline PRM shown in orange and our trained PRM in blue. Our trained PRM achieves a substantial performance improvement over the baseline.

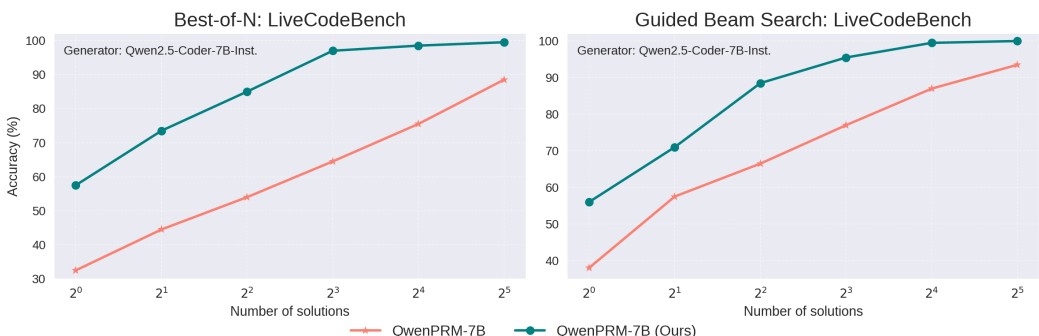

(b) In this figure, we compare Best-of-N alignment and Guided-Beam-Search alignment on LiveCodeBench task. The baseline PRM is shown in orange and our trained PRM is shown in blue. Our trained PRM achieves a substantial performance improvement over the baseline.

## A.4 PAIRWISE VS POINTWISE LOSS

**Setup.** Let a process reward model (PRM) produce a scalar score $s_\theta(y_t) \in \mathbb{R}$ for a step $y_t$. Let $Y^+ \sim \mathcal{D}_+$ denote correct steps and $Y^- \sim \mathcal{D}_-$ denote incorrect steps.

- Decision rule (thresholding): predict "correct" iff $s_\theta(y_t) \geq \tau$.
- False positive rate (FPR) at threshold $\tau$:
$$\mathrm{FPR}_\theta(\tau) = \Pr\left[s_\theta(Y^-) \geq \tau\right].$$
- True positive rate (TPR) at threshold $\tau$:
$$\mathrm{TPR}_\theta(\tau) = \Pr\left[s_\theta(Y^+) \geq \tau\right].$$

We now compare two training objectives that produce scores $s_{\theta_{\mathrm{pair}}}$ (pairwise BT) and $s_{\theta_{\mathrm{ce}}}$ (pointwise CE).

**Bradley–Terry objective.** The BT loss on pairs $(Y^+, Y^-)$ is the logistic loss on score differences:
$$\mathcal{L}_{\mathrm{BT}}(\theta) = \mathbb{E}_{Y^+, Y^-}\left[\log\left(1 + \exp(-(s_\theta(Y^+) - s_\theta(Y^-)))\right)\right].$$

Equivalently, it maximizes the likelihood that $Y^+$ outranks $Y^-$:
$$\Pr(Y^+ \succ Y^-) = \sigma\big(s_\theta(Y^+) - s_\theta(Y^-)\big).$$

Gradients are largest when a negative outranks a positive or the scores are too close, so BT explicitly pushes negatives down and positives up, widening margins.

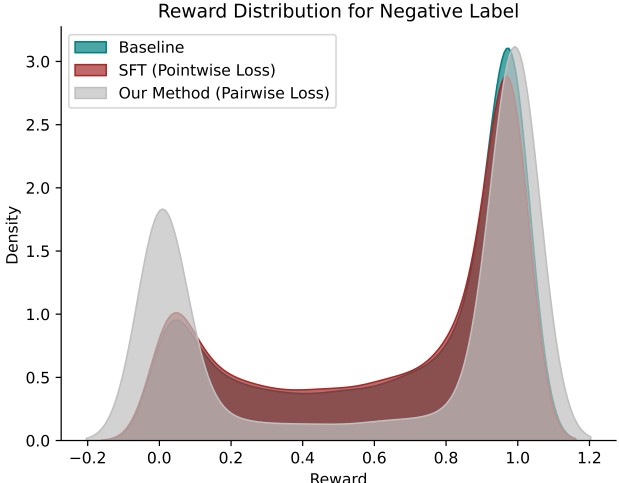

Figure 6: In this figure, we compare the reward for negative labels for the different QwenPRM models. In green, is the baseline QwenPRM. In red is the model trained using default cross-entropy loss and in silver is our method trained using pairwise loss. We observe that for our method the distribution around the negative reward increases as compared to the default PRM training method. Thus, showing that our method is better.

**Cross-entropy objective.** With binary labels $y_{GT} \in \{0, 1\}$ and $p_\theta(y_t) = \sigma(s_\theta(y_t))$, the CE loss is

$$\mathcal{L}_{\text{CE}}(\theta) = \mathbb{E}_Y[-\log s_\theta(Y)] + \mathbb{E}_Y[-\log(1 - s_\theta(Y))].$$

Gradients act independently, per example so there is no explicit coupling between positives and negatives, and CE does not directly optimize margins.

## A.5 TRAINING CURVES

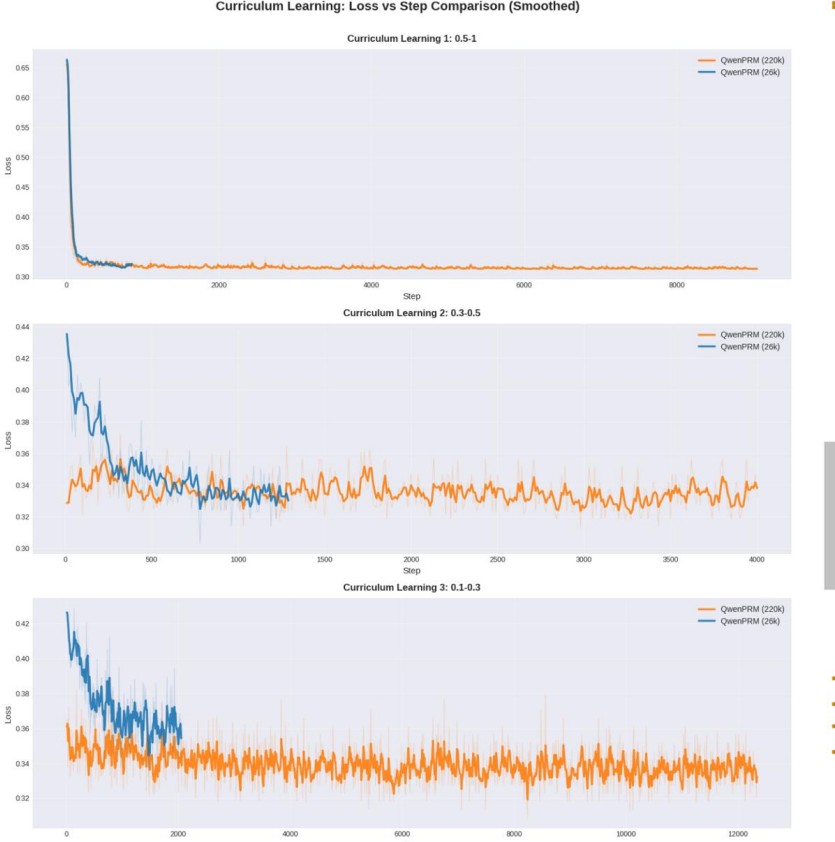

Figure 7: In this figure, we compare the loss curves for our data-augmented PRM training (220k datapoints) with baseline PRM training (26k datapoints). For the easier curriculum bins, both the baseline and our augmented method converge to similar loss values. However, as we move to the more difficult curriculum stages, our augmented method continues to reduce the loss as compared to the baseline. This indicates that in the easy regions, additional data offers limited benefit, whereas in the harder regions, our method benefits because of extra data.

## A.6 STATISTICAL PLOTS

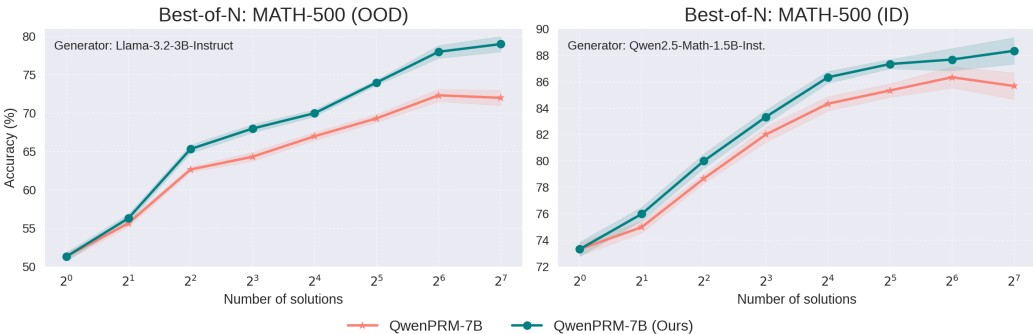

Figure 8: In this figure, we show the error plot which compare best-of-N alignment on MATH-500 using two different generator policies: LLaMA for the OOD policy (left) and Qwen for the ID policy (right). The baseline PRM is shown in orange and our trained PRM in blue. Across both settings, our trained PRM delivers a clear performance boost over the baseline, with the improvement being especially pronounced for the OOD policy.

## A.7 RESULTS ON REASONEVAL-7B PRM

Please see below results on another PRM model (ReasonEval-7B) to show generalizability of our approach. We see similar observations. As we move up in the curriculum learning round, our FPR decreases a lot with a slight increase in FNR. We also see continuous improvement in the PRMScore. This model was also reported in the Table 2 previously.

| Threshold / Curriculum | PRM Score | FPR | FNR |
|---|---|---|---|
| CL1 (0.5 − 1.0) | 61.0 | 78.79 | 4.21 |
| CL2 (0.3 − 0.5) | 61.8 | 69.76 | 8.05 |
| CL3 (0.1 − 0.3) | 62.3 | 66.8 | 9.9 |
| CL4 (0.0 − 0.1) | 62.3 | 61.7 | 13.6 |

Table 6: Performance of ReasonEval-7B PRM using our method to show generalizability of our approach.

## A.8 ABLATIONS ON CURRICULUM LEARNING BINS

We experimented with multiple threshold settings for curriculum learning and report the results in table 7 and 8. The final performance remains consistent across different bin configurations, which is expected given the gradual progression toward more confusing or difficult regions. Across all three curriculum-learning bin configurations, we observe a substantial decrease in the false-positive rate and only a small increase in the false-negative rate as we advance through the curriculum rounds.

| Threshold / Curriculum | PRM Score | FPR | FNR |
|---|---|---|---|
| CL1 (0.7 − 1.0) | 66.4 | 65.58 | 5.4 |
| CL2 (0.5 − 0.7) | 67.2 | 62.4 | 6.29 |
| CL3 (0.3 − 0.5) | 68.0 | 56.6 | 8.24 |
| CL4 (0.1 − 0.3) | 68.2 | 49.0 | 11.0 |

Table 7: Results on Qwen-PRM-7B training using our method with different curriculum learning bin.

| Threshold / Curriculum | PRM Score | FPR | FNR |
|---|---|---|---|
| CL1 (0.5 − 1.0) | 67.0 | 63.0 | 6.12 |
| CL2 (0.3 − 0.5) | 67.8 | 57.46 | 8.1 |
| CL3 (0.2 − 0.3) | 68.0 | 53.5 | 9.7 |
| CL4 (0.1 − 0.2) | 68.0 | 47.0 | 12.0 |

Table 8: Results on Qwen-PRM-7B training using our method with different curriculum learning bin.

