# OpenReview forum: "Cut the Overcredit: Precision First Process Rewards for Reasoning LLMs"
_ICLR.cc/2026/Conference — Submitted to ICLR 2026_

### Official Review · Reviewer_Wpkf · 2025-10-25

**Soundness:** 3
**Presentation:** 3
**Contribution:** 1
**Rating:** 4
**Confidence:** 3

**Summary:**

This paper discuss the critical issue of overcrediting (high false positive rate) in PRMs for reasoning LLMs. The authors provide both theoretical analysis and empirical evidence showing that false positives impose a ceiling on alignment performance. they propose a label-efficient approach: converting step annotations into positive-negative pairs, training with an Overcredit Contrastive loss, and applying lightweight augmentation with curriculum learning. Experiments demonstrate substantial reductions in false positive rates and improved F1.

**Strengths:**

1. Clear empirical improvement: The experiment section shows consistent downstream gains in guided search and Best-of-N
2. Good theoretical analysis: The FPR vs FNR asymmetry argument  is elegant and actionable
3. The Label-efficient method achieves substantial FPR reduction and improved macro F1

**Weaknesses:**

1.  The PRMs are known to be tricky and unstable in many settings, as reported in DeepSeek-R1. I would be more convincing to the community for direct comparison with GRPO (which avoids explicit PRMs). Showing that once some problem of PRMs is solved, it has clear potential to outperform ORM-driven methods. Currently, the paper only mentions some gain over existing SoTA PRMs, which somehow is not what people tend to use for baseline establishment.

2.  Would benefit from comparison with more PRM methods like implicit PRMs.

**Questions:**

None

---

> ### Author Response · Authors · 2025-11-21
> **Response to Reviewer (1/1)**
>
> We thank the reviewer for the thoughtful evaluation and for recognizing the strengths of our work, including the clear empirical gains in guided search and Best-of-N performance, the actionable theoretical analysis of the FPR–FNR asymmetry, and the effectiveness of our label-efficient approach in substantially reducing false positives and improving macro F1. We address the reviewer’s concerns in detail below.
>
> > (Weakness 1) The PRMs are known to be tricky and unstable in many settings, as reported in DeepSeek-R1. I would be more convincing to the community for direct comparison with GRPO (which avoids explicit PRMs). Showing that once some problem of PRMs is solved, it has clear potential to outperform ORM-driven methods. Currently, the paper only mentions some gain over existing SoTA PRMs, which somehow is not what people tend to use for baseline establishment.
>
> **Response to Weakness 1:**
>
> Yes, the SOTA reasoning algorithm GRPO uses an ORM (RLVR) instead of a PRM not because using PRM for RL training is fundamentally flawed, but because perfect (or high-quality) PRMs are unavailable. These neural models therefore suffer from reward hacking (which we address in this work) and they are computationally expensive to train, whereas RLVR evaluates only the final answer and avoids step-level supervision. However this also introduces well-known issues[1] such as sparse rewards, the zero-gradient problem, and slow convergence which can be mitigated using PRMs.
>
> - Our PRM is not in conflict with GRPO; rather, it can significantly strengthen GRPO when used in place of the traditional sparse reward, which suffers from the zero-gradient problem and slow convergence in GRPO, also mentioned in [2].
>
> - We present an analysis on PRMBench’s dataset to illustrate how our approach can improve GRPO. We analyze RLVR (the default reward used in GRPO) vs different PRM models.
>     - First, we count how often RLVR assigns positive labels to a trajectory.
>     - Next, we measure how many of these positively labeled trajectories contain incorrect intermediate steps. In GRPO training, such steps would still receive positive reward simply because the final answer is correct.
>     - We then evaluate how often a PRM can correctly identify these incorrect intermediate steps within RLVR’s positively labeled trajectories.
>     - Our trained PRM performs even better at detecting such errors, showing that this type of PRM can provide a denser and more reliable reward signal for GRPO training.
>
>
> |  | Positive Labels  | Negative Labels
> | -------- | -------- | -------- |
> | Rule-Based Reward (RLVR) |  5829 | 1140 |
> | Incorrect classifications by RLVR |  4886 | - |
> | Correct classifications by PRM given RLVR's incorrect positives (Baseline QwenPRM) |  3629 | - |
> | Correct classifications by PRM given RLVR’s incorrect positives (Our Trained QwenPRM) |  4326 | - |
>
>
> [1] Lightman, Hunter, et al. "Let's verify step by step." The Twelfth International Conference on Learning Representations. 2023.
> [2] Sun, Yiyou, et al. "RL Grokking Recipe: How Does RL Unlock and Transfer New Algorithms in LLMs?." arXiv preprint arXiv:2509.21016 (2025).
>
>
>
>
> > (Weakness 2) Would benefit from comparison with more PRM methods like implicit PRMs.
>
> **Response to Weakness 2:**
>
> We provide additional results below on a different PRM model (ReasonEval-7B) to show generalizability of our approach. We see similar observations as Qwen-PRM. As we move up in the curriculum learning round, our FPR decreases a lot with a slight increase in FNR. We also see continuous improvement in the PRMScore. This model was also reported in the Table 2 previously.
>
>
> | Threshold/Curriculum | PRM Score | FPR | FNR |
> | -------- | -------- | -------- | -------- |
> | CL1 (0.5 - 1.0)    |  61  |   78.79   | 4.21 |
> | CL2  (0.3 - 0.5)   |   61.8   |   69.76   |   8.05  |
> | CL3  (0.1 - 0.3)   |  62.3   |   66.8   | 9.9 |
> | CL4  (0.0 - 0.1)   |  62.3   |  61.7    | 13.6 |
>
>
> Can you suggest any specific implicit PRM literature you would like us to compare with?
>
>
> On comparing with a recent work on Implicit PRM [1], we observe our Qwen-7B PRM to be much higher in performance as compared to implicit PRM. The performance of [1] is in the range of PRM models: RLHFlow-8B-Mistral-Data and RLHFlow-8B-DS-Data, whereas Qwen-7B PRM is significantly better than RLHFlow-8B-Mistral-Data and RLHFlow-8B-DS-Data[2, 3].
>
>
>
>
> 1. Yuan, Lifan, et al. "Free process rewards without process labels." arXiv preprint arXiv:2412.01981 (2024).
> 2. Zhang, Zhenru, et al. "The lessons of developing process reward models in mathematical reasoning." arXiv preprint arXiv:2501.07301 (2025).
> 3. Song, Mingyang, et al. "PRMBench: A fine-grained and challenging benchmark for process-level reward models." arXiv preprint arXiv:2501.03124 (2025).

---

### Official Review · Reviewer_xBKo · 2025-10-28

**Soundness:** 3
**Presentation:** 3
**Contribution:** 3
**Rating:** 6
**Confidence:** 4

**Summary:**

This paper tackles the "overcredit" issue in PRMs, where incorrect reasoning steps receive high rewards, leading to a high rate of false positives that harms model alignment. The authors introduce a label-efficient solution using a novel Overcredit Contrastive (OC) loss, which trains the model on positive-negative pairs converted from existing annotations.

**Strengths:**

- This work introduces Over-credit Contrastive loss to address the false positive bias in Process Reward Models.
- This work proposes a label-efficient framework that utilizes existing data and reduces annotation overhead.
- This work successfully improves F1 score and robustness in experiments on PRMBench.

**Weaknesses:**

Overall, I acknowledge the motivation for this work. I have the following major concerns:
- The ablation study on the OCLoss is missing. It is unclear whether the performance gains stem from the data curation or the reward correction itself.
- The current reward appears to be polarizing. I am concerned this might train the model to be overly decisive (i.e., "black-or-white"), causing it to incorrectly assign extreme reward values to ambiguous or speculative reasoning steps. Therefore, I would like to see case studies that analyze these scenarios.
- The data efficiency appears to be low. Could you provide the learning curves for your proposed training method and compare them against a baseline method?
- Due to the stochastic nature of model sampling, the Best-of-N (BoN) experiments should report the mean and variance across multiple trials.
- Missing Reference:

[1] Towards Reasoning Era: A Survey of Long Chain-of-Thought for Reasoning Large Language Models

[2] From System 1 to System 2: A Survey of Reasoning Large Language Models

**Questions:**

- Could you provide an ablation study to disentangle the effects of the OCLoss from the data curation process? This would help clarify whether the observed performance gains are attributable to the reward correction mechanism or the data selection method.
- The reward function seems to favor decisive, binary outcomes. Have you analyzed whether this might inadvertently encourage the model to become overly decisive, especially in ambiguous scenarios where nuanced or speculative reasoning is required? A qualitative analysis of such cases would be valuable.
- Regarding training efficiency, could you share the learning curves for your proposed method and compare them against a baseline? This would provide clearer insight into the data efficiency and convergence properties of your approach.
- To account for the stochasticity of the sampling process in BoN experiments, could you report the mean and variance of the results over multiple independent trials?
- Add more references.

---

> ### Author Response · Authors · 2025-11-21
> **Response to reviewer (1/1)**
>
> We thank the reviewer for the thoughtful assessment and for highlighting the core strengths of our work:
>
> - (1) introducing the Overcredit Contrastive (OC) loss to address the false-positive bias in Process Reward Models;
> - (2) proposing a label-efficient framework that repurposes existing annotations to reduce supervision costs; and
> - (3) demonstrating improved F1 scores and robustness on PRMBench.
>
> We address the reviewer’s concerns in detail below.
>
>
> > (Weakness 1) Could you provide an ablation study to disentangle the effects of the OCLoss from the data curation process? This would help clarify whether the observed performance gains are attributable to the reward correction mechanism or the data selection method.
>
>
> **Response to Weakness 1:**
>
>
> To disentangle the effect of out OCLoss from the data-curation process, we compare our loss against the pointwise loss (the default cross-entropy loss used in PRM training) using the same curated dataset. Specifically, we fine-tune the same Qwen-PRM model using both the pairwise loss (ours) and the pointwise loss on the same set of positive and negative examples.
>
> We report the FPR comparison for both models below. Our results show that our pairwise OCLoss reduces false positives more effectively than the default pointwise loss. Additionally, we include the reward distribution plot comparing the two loss functions in Appendix Figure 6 (Section A.4).
>
>
> |  | False-Positive | False-Positive Rate |
> | -------- | -------- | -------- |
> | Baseline | 9109 | 69.34 |
> | SFT (Pointwise Loss) | 8836 | 67.26 |
> | Our Method (Pairwise Loss) | 8112 | 61.75  |
>
>
>
> > (Weakness 2) The reward function seems to favor decisive, binary outcomes. Have you analyzed whether this might inadvertently encourage the model to become overly decisive, especially in ambiguous scenarios where nuanced or speculative reasoning is required? A qualitative analysis of such cases would be valuable.
>
> **Response to Weakness 2:**
>
> We appreciate this concern, but we believe there may be a misunderstanding.
>
> In Best-of-N policy alignment, we sample n trajectories from the policy and compute a full trajectory score by aggregating step-level scores using the min or product operator. We then **compare these n trajectories and select the best one** based on their scores. This process **does not rely on rigid binary labels; it uses continuous PRM scores.** Binary labels are only used when evaluating the PRM’s classification performance, not during Best-of-N inference.
>
> **Empirically,** we do not observe the model becoming overly decisive. Instead, we consistently see improvements in precision and reduced FPR, while PRMBench’s “Sensitivity” and “Simplicity” metrics—which capture nuanced, intermediate reasoning behavior—remain stable or even improve. This shows that the model is not collapsing into binary judgments, but rather becoming better at distinguishing difficult negative steps.
>
> > (Weakness 3) Regarding training efficiency, could you share the learning curves for your proposed method and compare them against a baseline? This would provide clearer insight into the data efficiency and convergence properties of your approach.
>
> **Response to Weakness 3:**
>
> We have now added the training curves in the appendix (Figure 7). For the easier curriculum bins, both the baseline and our augmented method converge to similar loss values. However, as we move to the more difficult curriculum stages, our augmented method continues to reduce the loss as compared to the baseline. This indicates that in the easy regions, additional data offers limited benefit, whereas in the harder regions, our method benefits because of extra data.
>
> > (Weakness 4) To account for the stochasticity of the sampling process in BoN experiments, could you report the mean and variance of the results over multiple independent trials?
>
> **Response to Weakness 4:**
>
> We have now reported the mean–variance plot for the Best-of-N experiment on the Math-500, averaged over three runs. These plots are provided in updated paper Appendix section Figure 8.
>
> > (Weakness 5) Add more references.
>
> **Response to Weakness 5:**
>
> We have added the stated references in line 109.

---

### Official Review · Reviewer_zirL · 2025-10-29

**Soundness:** 3
**Presentation:** 2
**Contribution:** 3
**Rating:** 4
**Confidence:** 4

**Summary:**

This paper addresses reward hacking in PRMs for reasoning llm, where PRMs provide step-level supervision but often overcredit incorrect steps, leading to high false positive rates. The authors analytically demonstrate an asymmetry in BoN alignment: FPR imposes a precision ceiling that caps performance asymptotically, while false negative rates merely slow convergence.

**Strengths:**

1. The formal analysis of FPR vs. FNR asymmetry in BoN provides a rigorous, offering a clear rationale for prioritizing precision and filling a gap in prior PRM literature.
2. By repurposing existing datasets like PRM800K into 220K pairs via augmentation and curriculum, the method avoids costly new annotations, making it architecture-agnostic and easily integrable with discriminative or generative PRMs.
3. Substantial improvements in negative accuracy (e.g., from 30.66% to 50.89% on PRMBench) and downstream robustness (e.g., higher BoN accuracy on OOD policies) demonstrate real-world impact, with visualizations effectively illustrating reduced overcrediting.

**Weaknesses:**

1. The evaluation focuses primarily on math reasoning (e.g., MATH-500, AIME), with minimal exploration of broader domains like coding, potentially limiting generalizability despite claims of applicability to diverse reasoning tasks.
2. The thresholds in curriculum learning are handcrafted. There lacks ablation study on the sensitivity of such hyperparameters.
3. The compared baseline is merely Qwen-PRM. Other baselines in Table 2 are not compared.
4. The decrease in FPR comes at a cost of increase in FNR, showing that the method introduces a tradeoff.
5. The expansion strategy shows marginal gains, less than 1% improvements on PRMBench.
6. The method's reliance on a predefined difficulty curriculum (based on initial reward differences) may introduce sensitivity to baseline PRM quality; ablation studies on curriculum bins are underreported, risking instability in noisier datasets.
7. While analyzing credit assignment (min/product/sum), the paper assumes uniform step independence, overlooking potential correlations in real chains; this could undervalue compounding effects in long-horizon tasks beyond the tested benchmarks.

**Questions:**

- Is the proposed overcredit contrastive loss the vanilla Bradly-Terry loss? If so, why give it a new name?
- To mitigate the FPR issue, can we simply tune the threshold of PRM output rewards?
- The equations in Section 2 have wrong markers
- In Figure 3/4 captions, the orange and blue lines should be exchanged
- Why ThinkPRM has different scores in Figure 4 when n=1?

---

> ### Author Response · Authors · 2025-11-21
> **Response to Reviewer (1/3)**
>
> We thank the reviewer for the thoughtful evaluation and for highlighting the key strengths of our work, including the rigor of our **formal analysis on the FPR–FNR asymmetry** in Best-of-N alignment, the practicality of our **label-efficient framework that repurposes existing PRM datasets** without requiring new annotations, and the substantial **improvements in negative accuracy and downstream robustness demonstrated across PRMBench and OOD policies**. We address the reviewer’s concerns in detail below.
>
>
> > (Weakness 2) The thresholds in curriculum learning are handcrafted. There lacks ablation study on the sensitivity of such hyperparameters.
>
> **Response to Weakness 2:**
>
> We experimented with multiple threshold settings for curriculum learning and report the results below. The final performance remains consistent across different bin configurations, which is expected given the gradual progression toward more confusing or difficult regions. Across all three curriculum-learning bin configurations, we observe a substantial decrease in the false-positive rate and only a small increase in the false-negative rate as we advance through the curriculum rounds.
>
>
> Experiment 1
>
> | Threshold/Curriculum | PRM Score | FPR | FNR |
> | -------- | -------- | -------- | -------- |
> | CL1 (0.7 - 1.0)    |   66.4   |   65.58   | 5.4 |
> | CL2  (0.5 - 0.7)   |   67.2   |   62.4   | 6.29 |
> | CL3  (0.3 - 0.5)   |   68   |   56.6   | 8.24 |
> | CL4  (0.1 - 0.3)   |   68.2   |  49    | 11 |
>
>
> Experiment 2
>
> | Threshold/Curriculum | PRM Score | FPR | FNR |
> | -------- | -------- | -------- | -------- |
> | CL1 (0.5 - 1.0)    |   67  |   63   | 6.12 |
> | CL2  (0.3 - 0.5)   |    67.8  |   57.46   | 8.1 |
> | CL3  (0.2 - 0.3)   |   68  |   53.5   | 9.7 |
> | CL4  (0.1 - 0.2)   |   68   |  47    | 12 |
>
>
> > (Weakness 3) The compared baseline is merely Qwen-PRM. Other baselines in Table 2 are not compared.
>
> **Response to Weakness 3:**
>
> Please see below results on another PRM model (ReasonEval-7B) to show generalizability of our approach. We see similar observations. As we move up in the curriculum learning round, our FPR decreases a lot with a slight increase in FNR. We also see continuous improvement in the PRMScore. This model was also reported in the Table 2 previously.
>
>
> | Threshold/Curriculum | PRM Score | FPR | FNR |
> | -------- | -------- | -------- | -------- |
> | CL1 (0.5 - 1.0)    |  61  |   78.79   | 4.21 |
> | CL2  (0.3 - 0.5)   |   61.8   |   69.76   |   8.05  |
> | CL3  (0.1 - 0.3)   |  62.3   |   66.8   | 9.9 |
> | CL4  (0.0 - 0.1)   |  62.3   |  61.7    | 13.6 |
>
>
> > (Weakness 4) The decrease in FPR comes at a cost of increase in FNR, showing that the method introduces a tradeoff.
>
> **Response to Weakness 4:**
>
> A modest increase in FNR is expected when aggressively reducing FPR, but the key point is that the magnitude of improvement is asymmetric: **the reduction in FPR is far larger than the rise in FNR across all curricula.** This leads to a net gain in precision, negative accuracy, and downstream alignment.
>
> More importantly, our theoretical analysis in Section 2 shows that FPR and FNR have fundamentally different impacts on Best-of-N alignment: **high FPR imposes a hard asymptotic ceiling on accuracy, whereas FNR only slows convergence without affecting the final limit as $N \to \infty$**. Thus, improving FPR—even at the cost of a modest increase in FNR—is the correct direction of optimization for lifting the alignment ceiling in practical decoding and selection settings.
>
> > (Weakness 5) The expansion strategy shows marginal gains, less than 1% improvements on PRMBench.
>
> **Response to Weakness 5:**
>
> PRMBench’s [1] PRMScore is a **normalized, saturated composite metric**, not an accuracy measure where a 1% gain is minor. As shown in Song et al. (2025), even **frontier proprietary models** like GPT-4o and Gemini-2.0-Flash-Exp occupy a very narrow PRMScore band in the mid-60s, indicating that the **benchmark is intentionally high-resolution and near its upper bound.** To give a perspective there is <2% difference in the PRMScore of GPT-4o with GPT-mini and Gemini-2.0-flash-exp with Gemini-2.0-thinking-exp-1219.
>
> In this regime, small numerical changes in PRMScore correspond to **substantial underlying improvements—particularly in negative, positivie accuracy, FPR, FNR and model robustness.** In our case, the PRMScore movement may appear small, but it reflects large reductions in false positives and leads to clear, measurable gains in downstream alignment tasks.
>
> To get more prespective on this, please check Table 3 of PRMBench[1].
>
> [1] Song, Mingyang, et al. "PRMBench: A fine-grained and challenging benchmark for process-level reward models." arXiv preprint arXiv:2501.03124 (2025).

---

> ### Author Response · Authors · 2025-11-21
> **Response to Reviewer (2/3)**
>
> > (Weakness 6) The method's reliance on a predefined difficulty curriculum (based on initial reward differences) may introduce sensitivity to baseline PRM quality; ablation studies on curriculum bins are underreported, risking instability in noisier datasets.
>
> **Response to Weakness 6:**
>
> We experimented with both a predefined (static) curriculum and an adaptive (dynamic) curriculum, and consistently observed better performance with the predefined version. The comparative results are shown in the table below.
>
> Static (Predefined) Curriculum Results
>
> |  | PRM Score | FPR | FNR |
> | -------- | -------- | -------- | -------- |
> | CL-0    | 66.7     | 61.75 | 7.1 |
> | CL-1    | 67.2     | 56.8 | 9.15 |
> | CL-2    | 67.3     | 50.71 | 11.9 |
>
> Adaptive (dynamic) Curriculum Results
>
> |  | PRM Score | FPR | FNR |
> | -------- | -------- | -------- | -------- |
> | CL-0    |  66.7     | 61.75 | 7.1 |
> | CL-1    |  66.9     | 59.13 | 9.23 |
> | CL-2    |  66.7     | 55.07 | 12 |
>
>
> Static curricula have been extensively studied in the curriculum-learning literature [2, 3]. In fact works like [4] observes that adaptive curricula can oscillate in noisy domains, while static ones remain stable showing decept credibility to pre-defined curriculum method.
>
>
> Overall, our curriculum serves as a lightweight stabilizer, not a fragile dependency: the main improvements come from pairwise training itself, and the curriculum only helps in pushing the boundary further by ordering comparisons from easy to hard.
>
> We report **ablations** as part of Weakness 2 answer demonstrating that improvements are monotonic across bins and not driven by any single difficulty segment.
>
> [2] Bengio, Yoshua, et al. Curriculum learning. In Proceedings of the 26th Annual International Conference on Machine Learning (ICML '09). Association for Computing Machinery, New York, NY, USA, 41–48.
> [3] Weinshall, Daphna, et al. "Curriculum learning by transfer learning: Theory and experiments with deep networks." International conference on machine learning. PMLR, 2018.
> [4] Graves, Alex, et al. "Automated curriculum learning for neural networks." international conference on machine learning. Pmlr, 2017.
>
>
> > (Weakness 7) While analyzing credit assignment (min/product/sum), the paper assumes uniform step independence, overlooking potential correlations in real chains; this could undervalue compounding effects in long-horizon tasks beyond the tested benchmarks.
>
> **Response to Weakness 7:**
>
> We acknowledge that the analysis in Section 2 assumes step independence. This assumption simplifies the derivation, and in practice PRMs are trained with step-level labels that implicitly treat steps independently. To verify whether real chains exhibit strong correlations that could undermine the analysis, we compute:
> * $P_{err|err}$ : probability the next step is misclassified given the current step is misclassified.
> * $P_{err|ok}$ : probability the next step is misclassified given the current step is correct.
>
> Both the baseline PRM and our trained PRM show only a modest gap between these quantities, indicating weak error correlation and limited compounding beyond what the independence model predicts. The results are shown below:
>
>
>
> |  | P_{err/err} | P_{err/ok} | Total Steps |
> | -------- | -------- | -------- | -------- |
> | Baseline: Qwen-PRM     | 1684     | 1353     | 93942 |
> | Our Trained: Qwen-PRM on 226k data    | 1837     | 1437     | 93942 |
>
> Thus, while correlated errors do exist, they are not large enough to create destabilizing long-horizon compounding, and reducing FPR remains the dominant factor influencing downstream alignment.
>
>
> > (Weakness 1) The evaluation focuses primarily on math reasoning (e.g., MATH-500, AIME), with minimal exploration of broader domains like coding, potentially limiting generalizability despite claims of applicability to diverse reasoning tasks.
>
> **Response to Weakness 1:**
>
> We appreciate the suggestion. Results on LiveCodeBench (Best-of-N and Guided-Beam-Search) are now added in Appendix Figure 5b. We see consistent improvement in the coding task too.

---

> > ### Author Response · Authors · 2025-11-21
> > **Response to Reviewer (3/3)**
> >
> > > (Questions 1) Is the proposed overcredit contrastive loss the vanilla Bradly-Terry loss? If so, why give it a new name?
> >
> > **Response to Question 1:**
> >
> > Yes, our “Overcredit Contrastive Loss” is the negative log-likelihood of the Bradley–Terry model and is inspired by the pairwise preference loss used in reward model training for RLHF. We have now acknowledged this in line 261 (Section 3.1).
> >
> > However, there are certain challenges in directly applying the standard preference-tuning loss (from RLHF) to PRM training, and thats why we gave it a new name.
> >
> > 1. **Prefix-controlled comparisons.**
> > In OCLoss, only the final step differs between the positive and negative samples; the prefix (prompt + history of steps) is held fixed. In conventional preference modeling, the entire trajectory/answer differs, which changes the semantics of the BT comparison.
> > 2. **PRM score distribution vs ORM scalar reward.**
> > PRMs output a step-wise Bernoulli distribution over {correct, incorrect}, not a scalar reward. OCLoss integrates this *probabilistic* structure directly into the BT likelihood. ORM losses operate on scalar scores and do not model step-level distributions.
> >
> > Thus, while the underlying likelihood structure traces back to Bradley–Terry, the objective is tailored specific to PRM.
> >
> >
> > > (Questions 2) To mitigate the FPR issue, can we simply tune the threshold of PRM output rewards?
> >
> > **Response to Question 2:**
> >
> > To answer this question, it is important to clarify how policy alignment works and why tuning a threshold cannot fix the false-positive (FPR) issue. In downstream alignment methods such as Best-of-N or guided beam search, the model uses the raw PRM scores, not the binary labels obtained after thresholding. These methods compare trajectories by their actual scores, so the threshold plays no role in policy selection—it is only used when evaluating PRM classification performance.
> >
> > Our method improves alignment by teaching the PRM to rank trajectories correctly through a comparative (pairwise) loss, which reduces FPR at any fixed threshold and reduces policy misalignment, because the underlying score distribution becomes more separable.
> >
> > Crucially, changing the threshold merely shifts the operating point along the ROC curve; it does not change the relative ordering of positive and negative steps.
> >
> > > (Questions 3) The equations in Section 2 have wrong markers. In Figure 3/4 captions, the orange and blue lines should be exchanged
> >
> > **Response to Question 3:**
> >
> > We have addressed these minor edits. Please let us know if something is not addressed properly.
> >
> >
> > > (Questions 4) Why ThinkPRM has different scores in Figure 4 when n=1?
> >
> > **Response to Question 4:**
> >
> > Could you please clarify your concern?
> > To address the likely confusion:
> >
> > * The two plots (left and right) use different generator policies, so their $n=1$ scores need not match.
> > * Within each plot, the ThinkPRM and QwenPRM values differ because they are two distinct PRM models.

---

### Official Review · Reviewer_hFNL · 2025-10-31

**Soundness:** 2
**Presentation:** 2
**Contribution:** 2
**Rating:** 4
**Confidence:** 4

**Summary:**

This paper improves process reward models (PRMs) for reasoning-oriented language models, with a focus on mitigating false positives that cap alignment performance during guided decoding and Best-of-N (BoN) response selection. The authors provide an analytical argument explaining how false positives impose a precision ceiling in BoN settings, and propose a approach based on pairwise training using the Bradley–Terry loss. They augment existing step-level annotations with context-matched negative samples and use a curriculum over pair difficulty to further stabilize training. Experimental results on PRMBench and downstream alignment settings show improved precision, reduced false positive rates, and stronger alignment performance, especially for out-of-distribution generators.

**Strengths:**

- The paper presents a clear and useful analysis showing that false positives impose a hard ceiling on alignment, particularly in Best-of-N setups. The derivation is straightforward and helps justify the focus on reducing false positives.
- The proposed method is simple and effective, simply using Bradley-Terry loss leads to meaningful improvements in precision and reduced false positive rates.
- The experimental results are strong and consistent, demonstrating both improved PRM performance on PRMBench and better downstream alignment in guided beam search and Best-of-N selection, including for out-of-distribution policies.

**Weaknesses:**

- The method does not meaningfully address  label efficiency and data imbalance in supervision. The approach still depends on access to both positive and negative annotations at the step level, and similar data balancing could be achieved in pointwise training. The claims of label efficiency and data imbalance mitigation feel overstated.
- The proposed “Overcredit Contrastive Loss” is mathematically identical to the well-known Bradley–Terry (BT) loss, which has been used extensively in outcome reward modeling. Rebranding it without acknowledging the equivalence weakens the paper’s novelty. Bradley-Terry loss is mentioned in appendix but not main body, and I did not find the authors justifying the relation to BT loss.
- The writing could be tightened for conciseness and professionalism. For instance, the section around line 296–300 includes an unnecessary list that takes up space without adding clarity, making the paper feel less compact than it could be.
- While the paper frames reward hacking in the context of reinforcement learning, it does not include any RL experiments. This disconnects the theoretical claims from demonstrated practice and weakens the application to real-world RL training.

**Questions:**

see above

---

> ### Author Response · Authors · 2025-11-21
> **Response to reviewer (1/2)**
>
> We thank the reviewers for their thoughtful feedback and for highlighting the key strengths of our work:
>
> - (1) the clarity and utility of our theoretical analysis on false positives and their ceiling on alignment, especially in Best-of-N settings;
> - (2) the simplicity and effectiveness of our Bradley–Terry–style loss in reducing false positives and improving precision; and
> - (3) the strong experimental results.
>
> We address the concerns in details as follows.
>
>
>
> > (Weakness 1) The method does not meaningfully address label efficiency and data imbalance in supervision. The approach still depends on access to both positive and negative annotations at the step level, and similar data balancing could be achieved in pointwise training. The claims of label efficiency and data imbalance mitigation feel overstated.
>
> **Response to Weakness 1:**
>
> **Label Efficiency:** As clarified in Lines 276–279 (Section 3.2), our method is label-efficient because **it does not require explicit negative labels**. The data-augmentation strategy automatically induces negative counterparts: for any time step t, all future positive steps t’ > t are treated as negatives without manual annotation. Thus, only ~50% of the pairs require human annotation (the positives), while the remaining ~50%, all induced negatives, are free.
>
> **Data Imbalance in Pointwise vs Pairwise loss:** The reviewer is correct that class rebalancing can also be applied to pointwise (cross-entropy) training. However, pointwise loss trains a binary classifier that predicts the absolute correctness of an individual step (“is this step correct or not?”), whereas our pairwise loss trains a model to learn relative preference between two steps (“is step A better than step B?”). The latter directly optimizes ranking quality, which is what downstream alignment methods rely on, as reflected in the BON scores. The margin induced by the pairwise loss also leads to a clearer separation between positive and negative steps, which is crucial for reducing false positives.
>
> To show this we fine-tuned the same Qwen-PRM with both pairwise and pointwise loss on the same positive and negative data. We report in Table 1 the FPR comparison for both the models below.
>
> ***Takeaway*:** We observe that our method reduces false positives more than the pointwise loss which is the default loss. We also show reward distribution plot comparing pointwise and pairwise loss for training the PRM in appendix figure 6 (section A.4).
>
>
> **Table 1:** FPR comparison for pointwise vs pairwise trained models.
>
> |  | False-Positive | False-Positive Rate |
> | -------- | -------- | -------- |
> | Baseline | 9109 | 69.34 |
> | SFT (Pointwise Loss) | 8836 | 67.26 |
> | Our Method (Pairwise Loss) | **8112** | **61.75**  |
>
>
> As shown, pairwise training yields substantially lower FPR, exceeding the improvements achievable with pointwise balancing alone. We provide additional distributional comparisons in Appendix A.4.
>
>
> > (Weakness 2) The proposed “Overcredit Contrastive Loss” is mathematically identical to the well-known Bradley–Terry (BT) loss, which has been used extensively in outcome reward modeling. Rebranding it without acknowledging the equivalence weakens the paper’s novelty. Bradley-Terry loss is mentioned in appendix but not main body, and I did not find the authors justifying the relation to BT loss.
>
> **Response to Weakness 2:**
>
> Yes, our “Overcredit Contrastive Loss” is the negative log-likelihood of the Bradley–Terry model and is inspired by the pairwise preference loss used in reward model training for RLHF. We have now acknowledged this in line 261 (Section 3.1).
>
> However, there are certain challenges in directly applying the standard preference-tuning loss (from RLHF) to PRM training:
>
> 1. **Prefix-controlled comparisons.**
> In OCLoss, only the final step differs between the positive and negative samples; the prefix (prompt + history of steps) is held fixed. In conventional preference modeling, the entire trajectory/answer differs, which changes the semantics of the BT comparison.
> 2. **PRM score distribution vs ORM scalar reward.**
> PRMs output a step-wise Bernoulli distribution over {correct, incorrect} or {correct, incorrect, neutral}, not a scalar reward. OCLoss integrates this *probabilistic* structure directly into the BT likelihood. ORM losses operate on scalar scores and do not model step-level distributions.
>
> Thus, while the underlying likelihood structure traces back to the Bradley–Terry formulation, the objective is adapted to operate over distributional step-level outputs and prefix-controlled step comparisons.

---

> > ### Author Response · Authors · 2025-11-21
> > **Response to reviewer (1/2)**
> >
> > > (Weakness 3) The writing could be tightened for conciseness and professionalism. For instance, the section around line 296–300 includes an unnecessary list that takes up space without adding clarity, making the paper feel less compact than it could be.
> >
> > **Response to Weakness 3:**
> >
> > We appreciate the feedback. The manuscript has been polished and made compact throughout, and the list in Lines 296–300 has been removed from the main paper and moved to the appendix.
> >
> >
> > > (Weakness 4) While the paper frames reward hacking in the context of reinforcement learning, it does not include any RL experiments. This disconnects the theoretical claims from demonstrated practice and weakens the application to real-world RL training.
> >
> > **Response to Weakness 4:**
> >
> > Thank you for raising this point. Our primary research objective is to identify and correct **weaknesses in reward models** that lead to reward hacking, particularly the issues specific to process reward models. Our **method is intentionally designed to be general and applicable to any downstream policy optimization** approach, not just reinforcement learning. Accordingly, we evaluate the method across the most widely used alignment pipelines:
> > - Best-of-N Results: We show results for this most common policy alignment approach as part of Figure 4 in our paper.
> > - Reward Guided Beam Search Results: We show results for this policy alignment approach which is significantly affected by process supervision as part of Figure 3 in our paper.
> > - Analysis for GRPO: For a discussion of how our approach performs relative to GRPO, which uses rule-based RL-style optimization, please refer to our response to Reviewer Wpkf (Weakness 1).
> >
> > Across all these settings, we observe that reducing false positives consistently improves downstream alignment, exactly as predicted by our theoretical analysis. This demonstrates a direct and robust connection between the theory and practical alignment methods.
> >
> > In summary, our contribution is not framed as RL-specific, and none of our theoretical assumptions depend on reinforcement learning. Our method targets a general property of reward models that affects *all downstream optimization procedures*, including but not limited to RL based ones.

---

### Author Response · Authors · 2025-12-04
**Summary of the Rebuttal**

Dear AC,

We express our sincere gratitude to all reviewers for their thoughtful feedback and the time they dedicated to evaluating our work. Below, we provide a concise summary of the rebuttal phase to highlight both the consensus around the strengths of our contributions and the substantial improvements made to the paper in response to reviewer suggestions.

# 1. Consensus on Strengths

We are highly encouraged that the reviewers recognized the novelty, rigor, and significance of our paper:

### **Rigorous Theoretical Analysis**
Reviewers hFNL, zirL, and Wpkf commended our **theoretical analysis of the FPR–FNR asymmetry in Process Reward Models**, finding it rigorous, novel, and impactful for the community. They agreed that its implications for Best-of-N (BoN) alignment justify our focus on reducing false positives.

### **Innovative Methodology**

All reviewers praised our introduction of the **Overcredit Contrastive (OC) loss** for PRM training, noting its effectiveness in **reducing false positives** and its **label-efficient design**, which leverages existing annotations. Reviewers further highlighted the **substantial reduction in FPR** and **improved F1 scores** achieved by our method.

### **Extensive Evaluation**

Reviewers acknowledged the breadth and depth of our evaluation across mathematical reasoning tasks, including in-distribution and out-of-distribution policies, as well as multiple alignment schemes (Best-of-N and Guided Beam Search). They emphasized the **consistent improvements** observed across all scenarios.


# 2. Revisions & Responses

We thoroughly addressed every concern raised by the reviewers and conducted substantial additional experiments. A summary of revisions is provided below:

### **Reviewer hFNL:**

We added:
* Clarification on the **label-efficiency** of our method (Weakness 1).
* A detailed discussion of the **similarities and distinctions between OC Loss and Bradley–Terry Loss**.
* Additional experimental analysis for **RL-based alignment (GRPO)**, supplementing the BoN and Guided Beam Search results already in the paper.

### **Reviewer zirL:**

We incorporated:
* **LiveCodeBench coding results** (Appendix Fig. 5b).
* Results using an additional PRM base model (**ReasonEval-7B**) in Appendix A.7.
* **Curriculum learning ablations** (Appendix A.8).
* Clarifications on the **FPR–FNR tradeoff** (Weakness 4) and the **significance of a 1% PRM accuracy improvement** (Weakness 5).
* Experiments comparing **static vs. dynamic curriculum schedules** (Weakness 6).
* Analysis and discussion on the **step-independence assumption** and its implications (Weakness 7).

### **Reviewer xBKo:**

We added:
* A controlled comparison of **pairwise vs. pointwise losses using identical data**, isolating the effect of OC Loss from data curation.
* Clarification on why **reward model polarization does not affect downstream alignment**.
* **Training curves across curriculum rounds** (Appendix Fig. 7).
* **Statistical averaging over three runs** for Best-of-N experiments (Appendix Fig. 8).

### **Reviewer Wpkf:**

We added:
* Analysis showing how our **PRM improves GRPO relative to ORM**, particularly given the substantial reduction in false positives.
* Comparisons between our **PRM and recent works on implicit PRMs**.

---

### Meta-Review · Area_Chair_mjPk · 2026-01-07

**Summary:**

This paper improves process reward models (PRMs) for reasoning-oriented language models, with a focus on mitigating false positives that cap alignment performance during guided decoding and Best-of-N (BoN) response selection. They do so by using a pairwise BT loss on counterfactual negative completions they generate.

The paper lacks novelty -- right now it comes as a bag of tricks that is applied together without a lot of analysis or comparison or guidelines for how one should do it on their domain. Comparison to other design choices (e.g., other curricula or no curricula but designing data mixtures properly) or existing PRM training techniques are not present. Results are too similar to a baseline and don't seem to have confidence intervals, which is a problem since PRMBench numbers are too close. Beam search and policy TTS results are all done with very old models at this point, and in addition, these are not even the best models for reasoning in their version. No comparisons to state-of-the-art methods for RL-based training with PRMs. Overall, all of these seem very problematic for acceptance right now.

**Reviewer Concerns:**

The concern around overclaiming for the BT loss and LiveCodebench (coding evaluations) was addressed.

But concerns regarding RL experiments, other ways of constructing negatives, presence or absence of curricula, etc were not addressed.

**Reviewer Scores:**

I think the scores would have been the same (4, 4, 6, 4) or 4, 4, 6, 6. In either case, i don't see this paper crossing the borderline score.

---

### Decision · Program_Chairs · 2026-01-26

Reject